# Tailoring the Diameters of Electro-Mechanically Spun Fibers by Controlling Their Deborah Numbers

**DOI:** 10.3390/polym12061358

**Published:** 2020-06-17

**Authors:** Domingo R. Flores-Hernandez, Braulio Cardenas-Benitez, Sergio O. Martinez-Chapa, Jaime Bonilla-Rios

**Affiliations:** Escuela de Ingeniería y Ciencias, Tecnologico de Monterrey, Ave. Eugenio Garza Sada 2501, Monterrey 64849, NL, Mexico; drflores@tec.mx (D.R.F.-H.); braulio.cardenas@uci.edu (B.C.-B.); smart@tec.mx (S.O.M.-C.)

**Keywords:** near-field electrospinning, polymer nanofibers, polymer relaxation time, Deborah number, microelectromechanical systems, suspended fibers

## Abstract

Polymer solutions with different concentrations of SU-8 2002/poly(ethylene) glycol/tetrabutyl ammonium tetrafluoroborate (SU-8/PEO/TBATFB) were electrospun in a low-voltage near-field electrospinning platform (LVNFES) at different velocities. Their diameters were related to the concentration contents as well as to their Deborah (*De*) numbers, which describes the elasticity of the polymer solution under determined operating conditions. We found a direct correlation between the concentration of PEO/TBATFB, the *De* and the diameter of the fibers. Fibers with diameters as thin as 465 nm can be achieved for *De* ≈ 1.

## 1. Introduction

There are two well-known processes to obtain electrospun fibers, the near-field electrospinning (NFES) and the far-field electrospinning (FFES). NFES can be used to obtain micro and nanofibers when an electrostatic field is applied to a polymer solution or melt, as in the case of FFES. These micro and nanofibers have been used in applications, such as filtration, batteries, protective clothes, tissue engineering, controlled drug delivery, and, in recent years, sensors, actuators, optoelectronics, and microelectromechanical systems (MEMS) [1,2]. Moreover, NFES has been used to pattern photosensitive nanofibers that can be readily converted into carbon nanowires interconnections for carbon-MEMS sensor applications [3]. In this regard, the negative epoxy-based resins SU-8 series are commonly used as a carbon precursor, since they can be easily patterned into high aspect ratio three-dimensional (3-D) structures [4,5]. Additionally, properties, such as electrical and thermal conductivity of suspended carbon fibers produced with this technique and materials, have been recently studied, showing promising results for sensing and biosensing applications [6,7]. 

In its simplest form, the FFES setup produces meshes with randomly oriented fibers due to the bending instabilities of the polymer jet [8,9]. Nonetheless, for some applications, single-fiber deposition and/or alignment of fibers is crucial to have adequate performance for devices, such as field-effect transistors, actuators, nano-gap electrodes, optical and gas sensors, and other micro- and nano-devices [6]. The NFES technique uses short distances from the tip to the collector and a low voltage to avoid bending instabilities of the polymer jet to overcome these problems. Furthermore, a platform with three moving axes is used to create high-resolution patterns of fibers [10,11]. Even though the voltages used in NFES are smaller, as compared with FFES (1–2 kV), the operating electric field strength might cause an electric discharge that could degrade or damage samples when working with biological, pharmaceutical, or electronic applications [4]. In this work, we focused our attention on a low voltage near field electrospinning (LVNFES) that dramatically reduces the applied voltage and, as a consequence, the aforementioned disadvantages.

In both processes, LVNFES and FFES, several variables can affect the final properties of fibers and their accurate deposition. Process variables, such as flow rate, applied voltage, distance from tip to the collector, velocities of pattern fabrication, the conductivity of the solution, and others, have been studied in order to produce high-quality fibers [7,12,13]. Other aspects affecting the final diameter and morphology of fibers are the rheological properties of the polymer solution, which have been widely studied for FFES [14,15,16,17,18]. Although the literature regarding LVNFES and NFES has presented rheological data, none of these have combined the *De* with the process parameters in order to control the final morphology of the fibers [2,19].

Finding an analytically tractable solution that combines rheological data with the momentum balance equations during the electrospinning process is a formidable mathematical task. One strategy to attempt a solution involves transforming equations to their dimensionless form, followed by the application of a numerical method [20,21,22]. The dimensionless equations show a set of dimensionless numbers that relate the materials properties and processing conditions. In particular, the Deborah number (*De*) has been used by several authors in extensional flow [21,22,23,24], such as in the Giesekus model, which can be written as:(1)$$\tau_{zz}+De\left(v{\tau^{\prime }}_{zz}-2v^{\prime }\tau_{rr}\right)+\alpha \frac{De}{\eta_{p}}\tau^{2}{}_{zz}=2r_{\eta }v^{\prime }$$
where α is the mobility factor, *η*_p_ is the polymer solution viscosity, $\tau_{zz}$ and $\tau_{rr}$ are the axial and radial stresses, and $r_{\eta }$ is the ratio between the polymer and solvent viscosities, with the prime indicating derivatives respect to *z* [23]. A similar analysis can be done using the Phan-Thien-Tanner model (PTT) for uniaxial deformation. After a mathematical treatment performed by the authors of this work, it found a relation of the *De* and its relation with the fiber diameter:(2)$$d{\left(z\right)}^{2}=\frac{d_{0}^{2}\frac{dT\left(z\right)}{dc}\frac{v_{o}}{2\dot{\varepsilon }}}{1+T\left(z\right)\left(1-\frac{v_{0}Z}{2\dot{\varepsilon }DeL}\right)}$$
where $\frac{dT}{dc}$ is the change in the dimensionless axial stress over the change in the dimensionless distance, Z is entanglement destruction/formation rate, *L* is length in the z axis, $\dot{\varepsilon }$ is the extensional rate, and *De* is the Deborah number. Even though it is clear that there is an effect of *De* on the radius of the fibers, some parameters are difficult to measure or model for polymer diluted solutions. Therefore, a simpler way to analyze the effect of *De* on the fiber radius should be proposed.

In his work, Gadkari et al. [25] observed that the elastic stresses that were involved in FFES processes increase exponentially with the *De* and determined upper and lower limit values of *De* where the electrospinning is feasible. Carrol et. al. [21] numerically and experimentally investigated the effect of viscoelasticity by changing the polymer relaxation time (and therefore the *De*) for the FFES. The effect of the *De* against the radial stress and the axial stretching force was analyzed and Carrol et al. found that, as the jet moves from early stages, a very large extensional viscosity emerges making difficult for the electric forces to thin the polymer thread, resulting in thicker fibers. Gupta et. al. [24] studied Boger fluid for electrospinning. Boger fluids present a unique viscoelastic behavior, where the viscosity remains constant while the elasticity is a function of the deformation rate. Therefore, the analysis of pure elastic effects was achievable. The experimental results showed a strong direct correlation between the *De* and fiber diameters. The addition of polymer increased the relaxation times of the solutions and larger diameters were obtained. Nevertheless, all of these analyses have been conducted on FFES, but not in NFES or LVNFES. These results and an extensive literature research are compared in Table 1, which includes the used electrospinning technique, the polymer systems analyzed, ranges of voltages, concentrations, and velocities of deposition, and the effect of such parameters on the diameter of the fiber. Finally, Table 1 reveals the lack of investigation of the effect of the Deborah number for NFES, and specifically for LVNFES.

The effects that are caused by the electric parameters are reduced significantly and those due to the viscoelasticity of the solution prevail due to the low voltages used in LVNFES. A better understanding of the importance of the Deborah can be easily understood in Figure 1, where the effect of the deformation rate is observed for a non-Newtonian material, called silly putty^TM^. This material behaves as a viscous liquid at a slow deformation rate, but as an elastic solid at high deformation rates. For *De* < 0.5, the viscous effects prevail, which results in a thin diameter with a ductile-like failure. On the other hand, for *De* > 1, the elastic effects are dominant, giving thicker diameters and a brittle-like failure. 

In this work, the *De* is determined for a set of different polymer solutions, at different deposition rates for a modified Low-Voltage Near Field Electrospinning process that will be explained in Section 2.2, and it is named Contact Low Voltage Near Field Electrospinning (CLVNFES). The *De* numbers are calculated for each of the produced fibers using the characteristic relaxation times of each polymer solution, *λ_c_* (intrinsic to the nature of the material) at different fiber deposition times *t_c_* (dependent of the experimental conditions). The obtained fibers are characterized by Scanning Electron Microscopy (SEM) while using an image analysis software and their diameters correlated to their Deborah numbers. We demonstrate that by simply knowing the *λ_c_* of the polymer solution the deposition parameters can be adjusted in order to control the fiber diameter. It is expected that these results reduce the time of experimentation and lead the development of high resolution patterns made of polymer fibers, or even micro- or nano-3D printing applications.

## 2. Materials and Methods 

The polymer system consisted of a SU-8 2002 (SU-8)/poly(ethylene) oxide (PEO) solution with tetrabutylammonium tetrafluoroborate (TBATFB) as additive to increase the conductivity of the solution. SU-8 contains 71 wt% of cyclopentanone, that acts as the solvent of the system, see Table 2. SU-8 was obtained from MicroChem (Newton, MA, USA), while PEO and TBATFB of 99% purity were obtained from Sigma–Aldrich (Saint Louis, MI, USA). PEO has a viscosity-average molecular weight of Mv~4,000,000, with less than 1000 ppm of butylated hydroxytoluene (BHT)as an inhibitor. SU-8 is a high contrast, epoxy-based negative photoresist widely used for micromachining and microelectronic applications. All of the reactants were used as received.

The samples with higher PEO concentrations often required more stirring time to eliminate all of the PEO aggregates. All the samples were isolated from UV light to avoid any chance of cross-linking of SU-8 during the stirring step. The solution was extracted from the vial, with a 5 mL syringe, and then placed upside down for 24 h, also isolated from UV light, in order to eliminate bubbles from the solution.

### 2.1. Rheological Characterization of Polymer Solutions

All of the rheological tests were performed in a rotational rheometer (Physica MCR 301, Anton Paar, Graz, Austria) equipped with a cone-and-plate (CP) geometry (diameter of 24.98 mm, angle of 4.014°, and truncation of 249 µm). The experiments were conducted at 25 ± 0.1 °C and 24 h after polymer solution preparation. Frequency sweeps, in order to determine the loss and storage modulus, were performed at an amplitude strain of %_λ_ = 20 in the linear visco-elastic regime. The measurement chamber of the rheometer was saturated with cyclopentanone using a cotton ring that was positioned inside the chamber and around the lower plate, but avoiding contact with the cone and plate, to avoid solvent evaporation. Additionally, all of the samples were conditioned by applying a shear rate of 10 s^−1^ for two minutes followed by two minutes of rest due to the susceptibility of PEO solutions to any previous deformation.

### 2.2. Fiber Deposition Equipment

The TLVNFES setup, as in Figure 2a, comprises a three moving axis stage, a power supply (HVS448 3000 V, LabSmith, Livermore, CA, USA), a syringe pump (Pump 11 Elite, Harvard Apparatus, Cambridge, MA, USA)**,** and a 0.5 cc insulin syringe adapted with a chamfered syringe tip gauge 32 was used (Nordson EFD, Westlake, OH, USA). The components are controlled with a desktop computer. The depositions were conducted on SU-8 interdigitated electrode devices (IDEs) prepared according to the photolithography process guideline for SU-8 provided by MicroChem for SU-8 2015 with a height of 20 μm on a Si-SiO_2_ wafer, as reported elsewhere [5]. The programmed patterns included 30 parallel lines, with a separation of 50 μm and a length of 8500 μm. The gap between electrodes was *G* = 40 μm, see Figure 2b. The distance from the needle-to-substrate is *L* = 200 μm.

The NFES platform has three main parameters that need to be controlled to control the movements of the three-axis stage: (1) the velocity of fabrication (*v_f_*), which is the speed at which the fibers are deposited according to a pre-programmed pattern, (2) the inter-fabrication velocity (*v_i_*) required to relocate the moving stage to a subsequent fabrication pattern, and (3) the acceleration (*a*) to reach *v_f_* and *v_i_*, see Figure 2b. For all of the depositions, the inter-fabrication velocity and acceleration were constant and equal to *v_i_* = 5 mm·s^−1^ and *a* = 500 mm·s^−2^, while *v_f_* is variable. Given that the stretching and further deposition of the polymer threads occurs at *v_f_*, this variable becomes critical to understand the competition between viscous and elastic behavior of the polymer solution at different deformation rates.

The polymer flow rate was settled at 0.025 μL/min. A voltage of 100 V used was with the exception of the solution with 0.75 wt% and 1.00 wt% of PEO and TBATFB, where an initial voltage of 400 V was required to start the deposition, but was then immediately fixed to 100 V. Since the voltage was too low to eject a polymer jet, a modification to the typical NFES was done by contacting the tip of the needle with the substrate (identified with a letter B on Figure 3) to form a thread, which is then pulled up to a vertical distance of *L* = 200 μm. The fiber deposition can be performed once the polymer thread is formed. This is a modification to the LVNFES and renamed as Contact Low-Voltage Near Field Electrospinning (CLVNFES).

Three different set of experiments were conducted in order to determine the effect of concentration and velocity on fiber diameter: (A) fibers were deposited at several velocities (60, 40, and 20 mm·s^−1^) while using a constant formulation of 0.5%wt of PEO and TBATFB; (B) different formulations (1.00, 0.75, 0.50, and 0.25 wt%) were electrospun at 40 mm·s^−1^; and, (C) fibers were deposited at several velocities (10, 5, 1, 0.5 mm·s^−1^), using a constant formulation of 0.25%. Table 3 presents the experiment design.

Once the fibers were deposited, they were immediately exposed for 10 min. to a UV lamp (Blak-Ray B-100AP, Fisher Scientific, Hampton, NH, USA) at a wavelength of 365 nm to photopolymerize the SU-8. The morphology of the fibers was studied using a scanning electron microscope (SEM) by Zeiss EVO MA25 with an accelerating voltage of 10 kV after sputtering a 5 nm Au layer. Image processing software ImageJ by NIH was used to measure the fiber diameters from the SEM micrographs. For each sample, the fiber diameters were measured at 60 different points within the SEM image field.

## 3. Results and Discussion

### 3.1. Polymer Solution Relaxation Times

In order to determine the polymer critical relaxation times, *λ_c_*, we proposed a graphical method ease-of-use for an experimentalist who is not expert in rheology by avoiding the mathematical and numerical complexities that are involved in the viscoelastic deformation phenomenon. In this method, it is used the inverse value of the critical shear rate when a change in the slope of the viscosity curve take place. However, we also include the determination of *λ_c_*, using two alternative methods: the fitting by the Cross model, Equation (3), and by the mathematical relation between the relaxation spectra and the data obtained by oscillatory tests, Equation (4), each one with an increasing level of complexity.
(3)$$\eta =\frac{\eta_{0}}{1+\lambda_{c}{\dot{\gamma }}^{2/3}}$$
where *λ_c_* is a parameter that shows the onset of the non-Newtonian behavior, the $\eta_{0}$ is the apparent viscosity at zero shear rate, and $\dot{\gamma }$ is the shear rate. 

*λ_c_* it was also estimated using the values of the loss modulus, *G*’’(*ω*), from oscillatory tests, using a program that was developed by the authors to determine the relaxation spectra, *H*(*λ*). The data were further tweaked to improve exactitude of the Maxwell elements using the following equation
(4)$$G^{\prime \prime }\left(\omega \right)=\overset{N}{\underset{i=1}{\sum }}a_{i}\frac{\omega \lambda_{i}}{1+\omega^{2}\lambda_{i}^{2}}$$
where *a_i_* is the material elasticity constant of the *i*th Maxwell element. Since it is out of the scope of this work, the oscillatory tests, the code of the algorithm, and *η_i_*, *λ_i_*, and *a_i_*, are included in the Appendix A respectively.

Figure 4a summarizes, for all of the polymer solutions, the following information: (A) the response of the shear viscosity (*η*) as a function of shear rate ($\dot{\gamma }$), reported as markers. (B) The fitting of the Cross model, Equation (1), for shear viscosity, reported as solid lines, and (C) the graphical determination of the critical relaxation time (*λ_c_*), reported as dotted lines. Figure 4b shows the fitting of the G’’ while using Equation (4) where the relaxation spectrum is obtained, a logarithmic average of the relaxations times is calculated and reported in Table 4. 

We found that, for all solutions, *η* increased with the concentration of additives. Additionally, a shear-thickening behavior is observed at low shear rates until a critical value of $\dot{\gamma }$ ~ 0.3 s^−1^, after which a stabilization and a posterior shear thinning behavior occurs. The same behavior was observed by Bekkour [15] for PEO solutions in water for a wide range of concentrations. While the shear-thinning can be explained as a result of the alignment of the polymer chains to the flow due to the shear force applied, the shear-thickening is still not well understood. For the solution with fewer additives, PEO = TBATFB = 0.25 wt%, there is non-uniform shear-thinning behavior. The shear-thickening followed by a thinning change is present, but the stabilization of the solution with a Newtonian-like behavior is present from 0.1 s^−1^ to 10 s^−1^, where a second shear-thinning behavior appears, which indicates a more complex phenomenon occurring on the internal structure of the solution.

The *λ_c_*’s values are reported in Table 4 and they varied from 0.15 s to 0.68 s (graphical method) and from 0.02 to 0.31 s (using the Cross model), and from 0.21 to 1.37 (using the logarithmic average of H(λ)) for the solutions with concentrations from 0.25 wt% to 1 wt% of PEO and TBATFB, respectively. For all cases, the increase of *λ_c_* is proportional to the addition of both additives. This dependence between *λ_c_* and concentration is in agreement with previous reports where the relaxations times were determined for a wide number of PEO aqueous solutions with different molecular weight and concentrations [12,28]. 

### 3.2. Effect of Polymer Concentration and Speeds of Deposition on the Deborah Number

The Deborah number, *De*, was calculated using the logarithmic average relaxation times from H(λ). In its simplest form, *De* is defined as the ratio of the relaxation time of a determined material over the observation time, also known as characteristic time of the process, *De* = *λ_c_/t_d_*. For this case, *t_d_* was the time the fiber is stretched from the needle tip to the substrate, *t_d_* = *L/v_f_*, where *L* = 200 μm and *v_f_* is the velocity of fabrication.

The effect of the concentration of PEO and TBATFB on the diameter of the fibers, *d*, and the De, when *v_f_* = 40mm·s^−1^, is presented in Figure 5a. As it can be seen, as the concentration is decreased, it also decreases both *d* and *De*. Figure 5b shows the effect of *v_f_* on *d*, and the *De*, while the concentration was kept constant, [PEO] = [TBATFB] = 0.50 wt%. Similar behavior was observed, as *v_f_* was decreased the *De* and the diameters also decreased.

Additional experiments were performed using solutions with 0.25 wt% of the additives and a *v_f_* = 0.5, 1, 5, and 10 mm·s^−1^ since the effect of the reduction of concentration and velocity of deposition indicated a reduction of the diameters of the fibers. Table 5 summarizes the data from all of the fiber depositions; it includes the velocity of fabrication, concentration, the deposition time, the Deborah number, and the average diameter of the fiber. For the experiments, where *v_f_* is constant, the deposition time is constant and equal to *t_d_* = 0.005 s. In this case, when the concentrations decreased from 1.00 wt% of PEO and TBATFB to 0.25 wt%, the critical relaxation times, Deborah numbers, and diameters also decreased from *λ_c_* = 1.37 s to *λ_c_* = 0.21 s, *De* = 273.4 to *De* = 41.99, and *d* = 5.912 μm to *d* = 3.854 μm. When the concentration was kept constant at 0.5 wt% of PEO and TBATFB, the critical relaxation time was also constant, *λ_c_* = 0.74 s. The deposition times were 0.003 s, 0.005 s, 0.010, and 0.020 s for the velocities of fabrication of 60 mm·s^−1^, 40 mm·s^−1^, 20 mm·s^−1^, and 10 mm·s^−1^, respectively. The Deborah numbers were 221.12, 147.41, 73.71, and 36.85, while the diameters were 5.141 μm, 4.957 μm, 4.505 μm, and 4.198 μm. The only exception to this trend is the one where *v_f_* = 0.5 mm·s^−1^ and the concentration was 0.25 wt% of both additives, under these conditions the polymer thread looked stiffer, possibly due to the longer experimental times that the solvent had to evaporate. Additionally, experiments performed under those conditions usually presented problems such as tip clogging, polymer thread breakup and, therefore, regular depositions were hard to obtain. Nevertheless, a diameter reduction from *d* ≈ 5 μm (37 < *De* < 273), in the two first scenarios, to d ≈ 1 μm (*De* ≈ 1) in the improved one, stands for a reduction of 80% in diameter.

Our results suggested that the ability to control the diameter of the fibers could be attributed to the mechanism of the physical entanglement of the PEO long linear molecular chains. For the case where the concentrations were varied, it was obvious that the greater the concentrations of PEO increased the number of entanglements. These entanglements generate resistance to motion; therefore, the relaxation times increased, revealing an increment in the elastic response of the fluid. Similarly, for the cases where the concentration was constant, but the fabrication speed was varied, the uniaxial deformation was different. As the speed was increased, the material did not have enough time to dissipate the energy applied and respond in a more elastic way, leading to an increment in the diameters of the fibers. 

Figure 6 clearly shows a logarithmic dependence of *d* as function of *De*; this was expected due to the logarithmic nature of the relaxation times, from which *De* depends on. Moreover, this nature might explain why setups with *De* > 50 end with similar diameters around *d* ≈ 5 μm, while for 1 < *De* < 50, the diameters start to decrease significantly.

The effect of the velocity of deposition and polymer concentration has been analyzed for NFES with opposite results than the ones reported in this work [2,14,27,29]. However, their experiments also differ in the experimental conditions, while using longer distances from tip-to-collector, higher voltages, concentrations, and velocities of deposition, see Table 1. Under those conditions, the polymer solution is primarily affected by the electrostatic forces and the role of the viscoelasticity might be surpassed. In our set up, the potential difference was not even high enough to modify the droplet shape into the well-known Taylor cone. Instead, the voltage was merely used to promote the attach the polymer threads to the substrate through the electrostatic interaction. In these conditions, the forces due to viscoelasticity dominate the elongational process. Using a commercial USB HD Microscope 500x magnification by Shenzhen KSL Electronic Equipment Co., Ltd., in Figure 7, it portrayed the effect of the velocity of fabrication under low voltage and low velocities of fabrication. The fiber diameter increased as the *v_f_* was increased. By increasing the speed from *v_f_* = 1 mm·s^−1^ to *v_f_* = 3 mm·s^−1^ the diameter of the fiber before it was deposited increased from *d* = 6.9 μm to *d* = 18.8 μm. Finally, as the fiber got in contact with the post, an additional stretching reduced the diameter until the diameters reported in Table 5 were achieved.

Finally, Figure 8a presents a graphical comparison of all the experiments. The reduction of the additives leads to a decreasing diameter of the deposited fibers. Likewise, as *v_f_* is decreased, smaller fiber diameters are obtained. Figure 8b presents the SEM characterization of the fibers for the highest and lowest values of concentration or *v_f_*. The smallest fibers observed, *d* = 465 nm, were the result of combining the adequate concentrations and *v_f_* to obtain a *De* < 1. However, these small diameters were identified as isolated cases, not as reproducible as those reported in Table 5.

## 4. Conclusions

A series of LVNFES experiments were conducted on several SU-8 2002 solutions with concentrations of PEO and TBATFB that ranged from 0.25–1.00 wt% as additives, in order to determine the effect of their rheological properties on the diameters of fibers deposited between interdigitated electrodes. Using the Cross model for *η*, the measured shear viscosity data of the samples were fitted to determine the critical relaxation times of the solutions. Our results indicated that the presence of additives directly affected the viscosity of the samples, with higher concentrations yielding higher viscosities and relaxation times, ultimately leading to thicker fiber diameters. These observations can be explained by the increase of the number of entanglements in the solutions, which affect their viscoelastic response upon deposition. Likewise, the velocity of deposition was modified from 60 mm·s^−1^ to 1 mm·s^−1^ to work along with different uniaxial deformation rates. As the velocity and concentrations were reduced, the material had more time to dissipate the induced stresses and the reduction of entanglements allows for the material to be stretched easier; in consequence, the fiber diameter decreased. Further analysis of the transition of the liquid-like to solid-like behavior of these polymer solutions was performed by observing the Debora number. We found that the lower the De, the smaller the diameter of the fiber. Since the *De* is defined as the ratio of the polymer relaxation time over the characteristic time of the process, *De* = *λ_c_/t_d_*, as the values decreased and approach to *De* = 1, the characteristic time of the process becomes long enough to allow the material to relax, dissipate the tension forces, and start to behave in a liquid-like manner, inducing a reduction of the polymer thread diameter. Most importantly, our results demonstrate a cost-effective method to select the optimal process variables or polymer formulations that allow for tailoring the fiber diameters for contact LVNES setups since the Deborah number can be used to describe the viscoelastic behavior of any material.

## Figures and Tables

**Figure 1 polymers-12-01358-f001:**
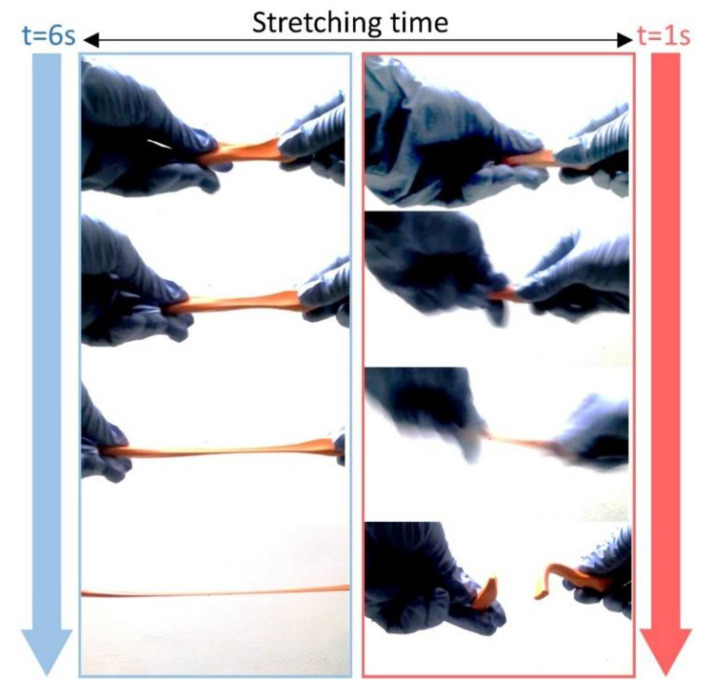
Effect of different strain rates on the diameter of the Silly Putty sample, a non-Newtonian fluid.

**Figure 2 polymers-12-01358-f002:**
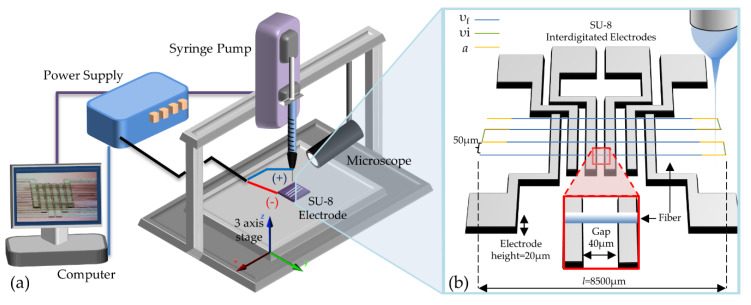
(**a**) Schematic of Contact Low-Voltage NFES experimental setup and (**b**) patterned deposition of fibers on SU-8 electrodes; velocities of fabrication, *v_f_* in blue, inter-fabrication, *v_i_* in green, and the acceleration, *a* in yellow, are schematically represented.

**Figure 3 polymers-12-01358-f003:**
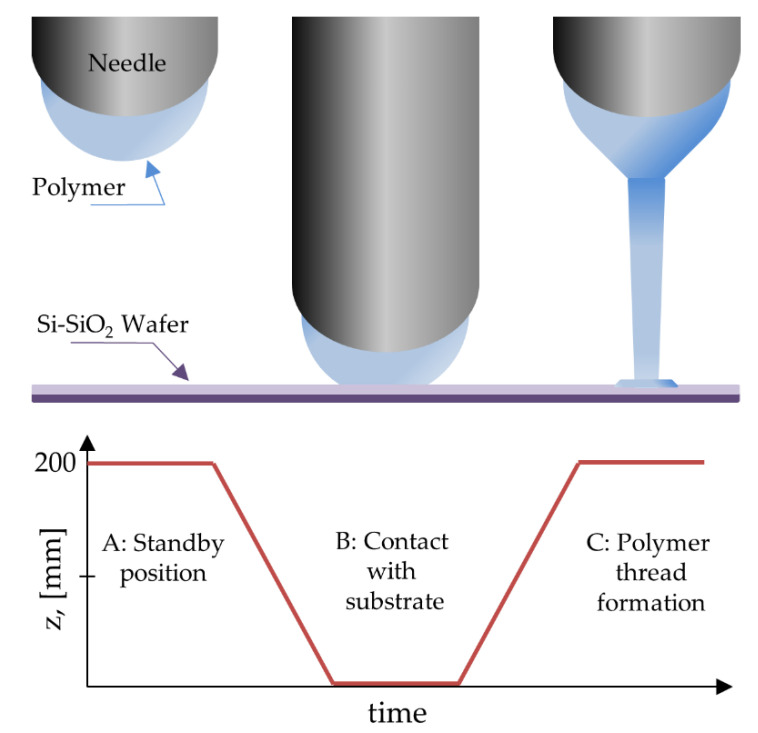
Polymer thread initiation by Contact Low-Voltage Near-Field Electrospinning.

**Figure 4 polymers-12-01358-f004:**
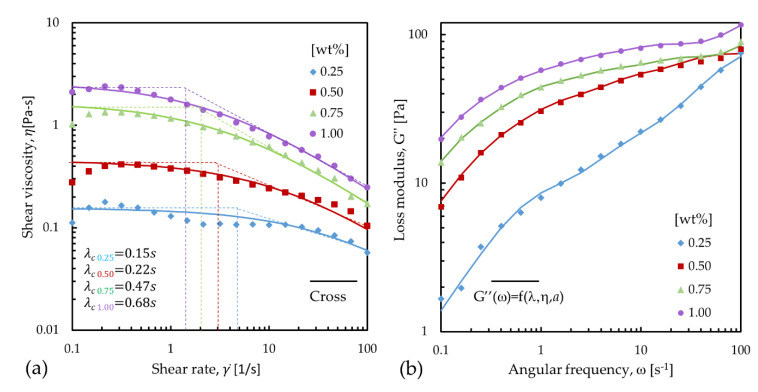
(**a**) Flow curves of SU-8 2002 solutions with concentrations from 0.25 wt% to 1.00 wt% of poly(ethylene) oxide (PEO) and tetrabutylammonium tetrafluoroborate (TBATFB). Graphical determination of the relaxation times using the data fitted from the Cross model. (**b**) Fitting of the G’’ as function of λ, η, and *a* using Equation (4).

**Figure 5 polymers-12-01358-f005:**
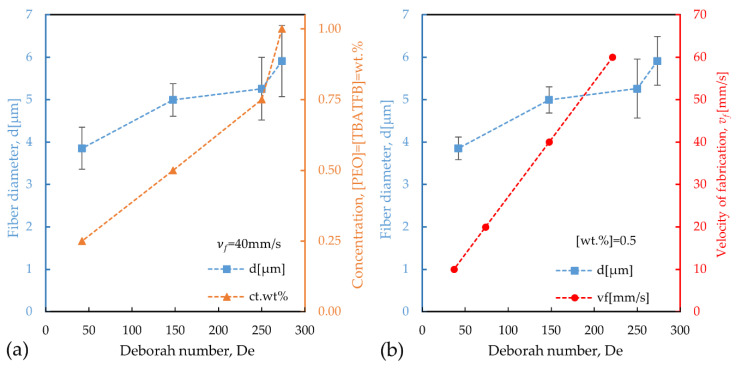
(**a**) Effect of the concentration of PEO and TBATFB on the Deborah number and the diameter of the fibers when *v_f_* = 40 mm/s. (**b**) Effect of *v_f_* on the Deborah number and diameter of the fibers for a concentration of 0.50 wt% of PEO and TBATFB.

**Figure 6 polymers-12-01358-f006:**
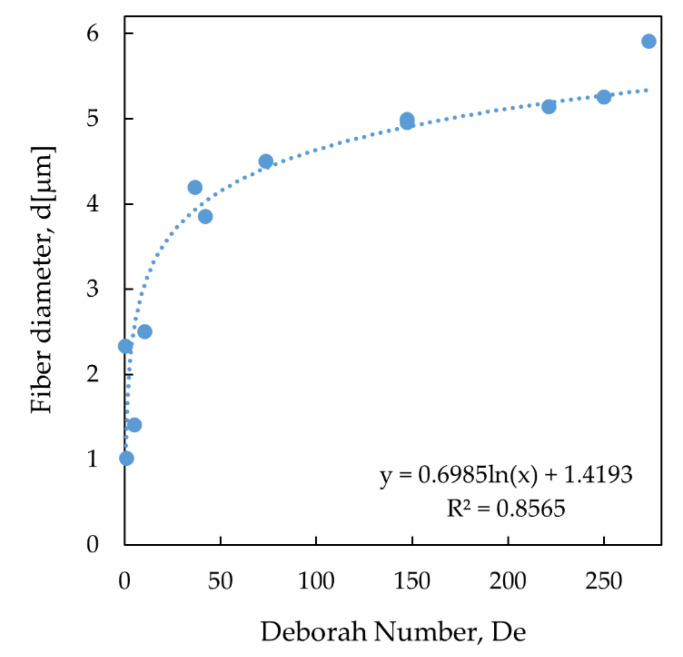
Diameter of the fibers as function of the Deborah number.

**Figure 7 polymers-12-01358-f007:**
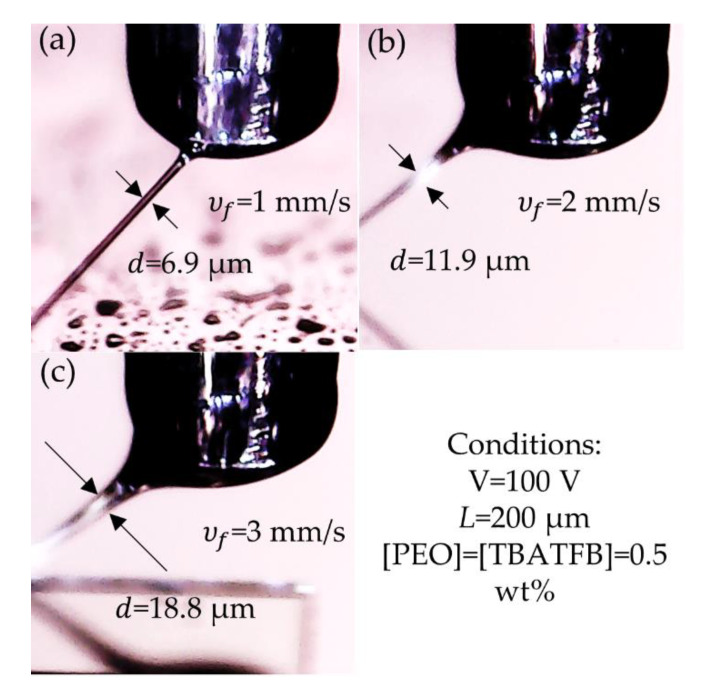
Effect of the velocity of the stage on the diameter of the polymer jet: (**a**) *v_f_* = 1 mm·s^−1^, (**b**) *v_f_* = 2 mm·s^−1^, and (**c**) *v_f_* = 3 mm·s^−1^.

**Figure 8 polymers-12-01358-f008:**
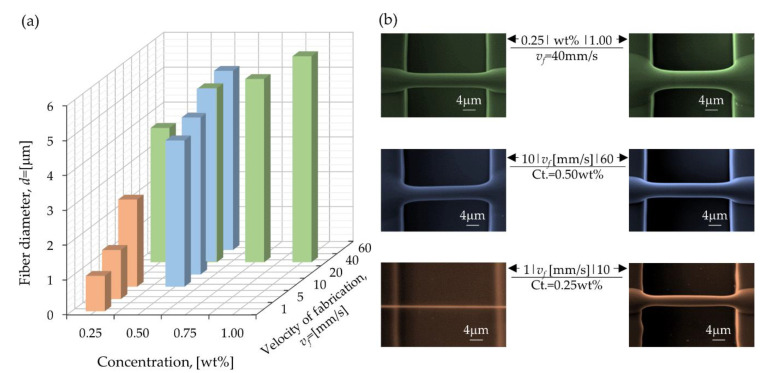
(**a**) Average diameter of fibers for all the experiments varying velocities of fabrication and concentrations of [PEO] = [TBATFB]. (**b**) SEM images of the fibers for the highest and lowest values of concentrations or velocities.

**Table 1 polymers-12-01358-t001:** Low-voltage near-field electrospinning platform (LVNFES), near-field electrospinning (NFES), and far-field electrospinning (FFES) operation.

Technique	Polymer/Solvent/Additive	Voltage, V [kV]	Deposition Speed, v_f_ [mm/s]	Concentration [wt%]	Fibers Diameter, *d* [µm]	Velocity-Diameter Effect	Concentration-Diameter Effect	Deborah Number, *De*	Ref.
LVNFES	PEO/DI Water/-	0.2–0.4	20–40	1–3	0.05–0.425	d∝1/v	d∝c	Correlates the relaxation time with the entanglement of the polymer network, but provides no further analysis	[2]
LVNFES	PEO/DIW/-	1.5	0–6.6	3–8	-	-	-	-	[24]
LVNFES	PEO/Water/-	0.05–0.15	100	3–10	0.1–0.3	-	d∝c	-	[9]
	PVP/EtOh/-	0.05–0.20		15–25	1–2				
	PS/DMF/-	0.05–0.70		20–30	0.5–1.2				
	Gelatin/HOAc-EtOAc/-	0.10–0.15		19	-				
LVNFES	PEO/SU-8 2002/TBATFB	0.1	1–60	0.25–1.00	0.976–7	d∝v	d∝c	d∝*De* in Contact-LVNFES	This work
HCNFES	PVDF/Dimethyl sulfoxide/ZONYL and acetone	10-16	942–1989	16–20	0.1–1.2	d∝1/v	N/A	-	[25]
NFES	PEO/DIW/-	1	50–250	3–5	0.3	-	d∝c	-	[26]
NFES	PEO/DI W/-	0.6-1.2	120	03–09	0.04-2.3	N/A	d∝c	-	[12]
NFES	PCL/Chloroform and N,N-dimethyl-formamide/none	2	20–45	10	6–32	d∝1/v	N/A	-	[27]
NFES	PEO/DIW/EtOH	1.5	3.82–8.59	4	1.75–4.76	d∝1/v	N/A	De, is used within a momentum balance equation. De effects on the	[21]
FFES	PIB/PB	9-13	N/A	2k,4k,8k ppm	0.08–0.30	N/A	d∝c	d∝*De* for FFES	[19]
FFES	PEO/DIW/PEG	-	N/A	0.1–0.2	2.7–9.5	N/A	d∝c	De and its influence over electrospinning process is deeply studied for FFES. They Define critical De values to obtain uniform fibers.	[20]
FFES	LMwPVA/DIW/HMwPVA	17-32	N/A	26–34	0.066–0.325	N/A	d∝c	d∝*De* in FFES	[22]

^1^ N/A stands for Does Not Apply; ^2^ "-" Stands for not reported.

**Table 2 polymers-12-01358-t002:** Prepared polymer solutions.

Solution	Concentration [wt%]		
	SU-8	PEO	TBATFB
1	99.50	0.25	0.25
2	99.00	0.50	0.50
3	98.50	0.75	0.75
4	98.00	1.00	1.00

**Table 3 polymers-12-01358-t003:** Set of experiments.

Concentration [wt%]				Velocity of Deposition [mm/s]						
Solution	SU-8	PEO	TBATFB	60	40	20	10	5	1	0.5
1	99.5	0.25	0.25		A		C	C	C	C
2	99.0	0.50	0.50	B	A/B	B				
3	98.5	0.75	0.75		A					
4	98.0	1.00	1.00		A					

**Table 4 polymers-12-01358-t004:** Critical relaxation times obtained graphically, with the Cross model, and with the relaxation spectra.

Solution	Relaxation Time, *λ_c_* [s]		
	Cross	Graphical	H(λ)
0.25	0.02	0.15	0.21
0.50	0.07	0.22	0.74
0.75	0.25	0.47	1.25
1.00	0.31	0.68	1.37

**Table 5 polymers-12-01358-t005:** Summary of fiber depositions.

Velocity of Fabrication, *υ_f_* [mm/s]	Concentration [PEO = TBATF]=wt%	Critical Relaxation Time, *λc* [s]	Deposition Time, t_d_ [s]	Deborah Number, *De*	Average Diameter of Fibers, *d* [µm]	Std. Dev., [µm]
40	1.00	1.37	0.005	273.44	5.912	0.839
40	0.75	1.25	0.005	249.95	5.259	0.742
40	0.50	0.74	0.005	147.41	4.992	0.386
40	0.25	0.21	0.005	41.99	3.854	0.493
60	0.50	0.74	0.003	221.12	5.141	0.577
40	0.50	0.74	0.005	147.41	4.957	0.694
20	0.50	0.74	0.010	73.71	4.505	0.305
10	0.50	0.74	0.020	36.85	4.198	0.267
10	0.25	0.21	0.020	10.50	2.502	0.301
5	0.25	0.21	0.040	5.25	1.406	0.184
1	0.25	0.21	0.200	1.05	1.016	0.153
0.5	0.25	0.21	0.400	0.52	2.331	0.351

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
