# Peer review of "Tailoring the Diameters of Electro-Mechanically Spun Fibers by Controlling Their Deborah Numbers"

_polymers, 2020, doi:10.3390/polym12061358_

Round 1

Reviewer 1 Report

Dear authors,

the reviewed version of the manuscript took into consideration the reviewers comment on the first version.

The paper deserves publication.

Two minor things should be amended/added.

1.I guess that the oscillatory rheological tests were carried out at small straiin amplitued, in the  linear viscoelasticity regime. However it is not reported in the text. I'd suggest to add this information, along with the imposed strain amplitude: this could be done adding a line in the section: 2.1 Rheological Characterization of Polymer Solutions.

2. In eq 4 (Maxwell model applied to G'') the term ai is not representi the viscosity of each viscous element of the Maxwell analogical model, but the elastic constant of the elastic element of the Maxwell model

Reviewer 2 Report

The authors have well responded to my comments. I recommend publication.

Author Response

We appreciate the reviewer for commentaries. We are glad to know that there are no more pendant revisions.

This manuscript is a resubmission of an earlier submission. The following is a list of the peer review reports and author responses from that submission.

Round 1

Reviewer 1 Report

See the attached pdf file

Reviewer 2 Report

Flores-Hernández et al. in this work correlate the fiber diameter and the Deborah number in a low-voltage near-field electrospinning platform (LVNFES). The Deborah number was controlled by the polymer concentration and the deposition velocity. They found that as the velocity and the concentrations were reduced, the fiber diameter decreased. These results can provide a good guideline for the LVNFES experimentation. I recommend publication of the manuscript after the following comments are addressed.

It is stated that SU-8 2002 was used for fabricating high-performance micro and nano-sensors. But it is not clear to me why this is necessary in this work. This should be further explained. The solvent used for the electrospinning should be clearly noted in the manuscript. It is stated that the velocity of fabrication (vf) is the speed at which the fibers are deposited according to a pre-programmed pattern. This is clear enough. Since vf is the most important parameter in this work, the definition of vf should be more understandable What is the reason for “as the De approached values near 1, the diameters of the fibers were significantly reduced?” This should be further elaborated. There are many typos in the manuscript.